# Antithrombotic Effects of Fostamatinib in Combination with Conventional Antiplatelet Drugs

**DOI:** 10.3390/ijms23136982

**Published:** 2022-06-23

**Authors:** Maan H. Harbi, Christopher W. Smith, Fawaz O. Alenazy, Phillip L. R. Nicolson, Alok Tiwari, Steve P. Watson, Mark R. Thomas

**Affiliations:** 1Institute of Cardiovascular Sciences, College of Medical and Dental Sciences, University of Birmingham, Birmingham B15 2TT, UK; mhh887@student.bham.ac.uk (M.H.H.); c.smith.14@bham.ac.uk (C.W.S.); faenzy@ju.edu.sa (F.O.A.); p.nicolson@bham.ac.uk (P.L.R.N.); s.p.watson@bham.ac.uk (S.P.W.); 2Pharmacology and Toxicology Department, College of Pharmacy, Umm Al-Qura University, Makkah 24381, Saudi Arabia; 3Department of Vascular Surgery, University Hospitals Birmingham NHS Foundation Trust, Birmingham B15 2GW, UK; alok.tiwari@uhb.nhs.uk; 4Department of Cardiology, University Hospitals Birmingham NHS Foundation Trust, Birmingham B15 2GW, UK

**Keywords:** fostamatinib, R406, Syk, tyrosine kinase, antiplatelet therapy, antithrombotic therapy, arterial thrombosis

## Abstract

New antithrombotic medications with less effect on haemostasis are needed for the long-term treatment of acute coronary syndromes (ACS). The platelet receptor glycoprotein VI (GPVI) is critical in atherothrombosis, mediating platelet activation at atherosclerotic plaque. The inhibition of spleen tyrosine kinase (Syk) has been shown to block GPVI-mediated platelet function. The aim of our study was to investigate if the Syk inhibitor fostamatinib could be repurposed as an antiplatelet drug, either alone or in combination with conventional antiplatelet therapy. The effect of the active metabolite of fostamatinib (R406) was assessed on platelet activation and function induced by atherosclerotic plaque and a range of agonists in the presence and absence of the commonly used antiplatelet agents aspirin and ticagrelor. The effects were determined ex vivo using blood from healthy volunteers and aspirin- and ticagrelor-treated patients with ACS. Fostamatinib was also assessed in murine models of thrombosis. R406 mildly inhibited platelet responses induced by atherosclerotic plaque homogenate, likely due to GPVI inhibition. The anti-GPVI effects of R406 were amplified by the commonly-used antiplatelet medications aspirin and ticagrelor; however, the effects of R406 were concentration-dependent and diminished in the presence of plasma proteins, which may explain why fostamatinib did not significantly inhibit thrombosis in murine models. For the first time, we demonstrate that the Syk inhibitor R406 provides mild inhibition of platelet responses induced by atherosclerotic plaque and that this is mildly amplified by aspirin and ticagrelor.

## 1. Introduction

Antiplatelet therapy is the cornerstone of the treatment strategy for acute coronary syndrome (ACS) patients. Platelet-mediated thrombus formation after atherosclerotic plaque rupture is the critical step that drives the underlying pathophysiology. Dual antiplatelet therapy (DAPT), consisting of aspirin and a potent P2Y_12_ inhibitor (such as ticagrelor or prasugrel) is recommended to reduce the risk of recurrent atherothrombotic events. However, the antithrombotic benefit of DAPT is at the expense of an increased annual risk of major bleeding (approximately 2–4%) [1,2,3]. After one year of DAPT, patients with ACS are treated with aspirin alone as the bleeding risk at this stage with P2Y_12_ inhibitors generally outweighs the antithrombotic benefit [1]. Therefore, there is minimal scope to introduce new antithrombotic drugs unless they have a minimal effect on bleeding. More effective antithrombotic strategies are required, however, as approximately 10% of patients still present with atherothrombotic events in the first year after ACS despite DAPT, and about 10% of patients have a recurrent event between 1 and 4 years if treated with aspirin alone [2,3,4].

The platelet receptor glycoprotein VI (GPVI) has been identified as a key receptor involved in atherothrombosis [5,6]. Exclusive to the platelet lineage, GPVI binds to collagen and fibrin and triggers powerful platelet activation via Src family kinases (SFKs), spleen tyrosine kinase (Syk), and Tec family kinases [7,8,9,10,11]. Syk is also involved in platelet activation via hemi-immunoreceptor tyrosine–based activation motif (hemITAM) (hemITAM)-containing receptor CLEC-2 and ITAM containing Fc*γ*RIIA, as well as the von Willebrand factor (VWF) receptor GPIb-IX-V and integrin αIIbβ3 [12,13,14,15]. The inhibition of Syk in mice has been shown to protect against arterial thrombosis without altering bleeding times, suggesting it does not affect haemostasis [16,17,18,19,20].

The oral prodrug fostamatinib (active metabolite, R406) is a first-in-class Syk inhibitor approved in the United States for the treatment of immune thrombocytopenia (ITP) [21]. To date, fostamatinib remains the only Syk inhibitor approved for clinical use. For in vitro studies, we used the active metabolite R406 in a concentration range relevant to the reported Cmax achieved with a recommended dose of 200 mg/day (395–1140 ng/mL or 0.8–2.5 µM) [22]. In this study, we investigated whether fostamatinib could be repurposed as a novel antithrombotic drug and, for the first time, assessed its effects on atherosclerotic plaque-mediated platelet responses alone and in combination with conventional antiplatelet drugs used for the treatment of atherothrombosis.

## 2. Results

### 2.1. R406 Inhibits Atherosclerotic Plaque-Induced Platelet Signalling

As fostamatinib is a prodrug, which requires metabolising to become active, its active metabolite R406 was used to study its effects in vitro. To determine the effect of R406 on GPVI signalling, the phosphorylation of downstream kinases was assessed in washed platelets stimulated with atherosclerotic plaque homogenate (Figure 1). The tyrosine phosphorylation of Syk, linker for activation of T cells (LAT), and PLC*γ*2 was abolished by 10 µM of R406. The effects were concentration-dependent, as 1 µM of R406 only had a marginal effect on Syk kinase but not on other kinases downstream of Syk. R406 at a 10 µM concentration also blocked upstream Src Y418 autophosphorylation (Figure 1B). Similar results were observed with platelets stimulated with collagen and the GPVI-specific agonist collagen-related peptide (CRP) (Appendix A).

### 2.2. R406 Reduces Platelet Adhesion on Atherosclerotic Plaque under Static Conditions

Syk has previously been shown to have a key role in GPVI- and αIIbβ3-mediated platelet spreading [15,23]. Therefore, we investigated the effect of R406 on platelet spreading and adhesion on atherosclerotic plaque homogenate, collagen, fibrin, and fibrinogen-coated coverslips (Figure 2). We also investigated whether R406 had a synergistic effect on platelet spreading when combined with aspirin.

R406, at a concentration of 10 µM, reduced platelet adhesion on atherosclerotic plaque under static conditions but did not affect the platelet surface area. Similarly, aspirin and/or ticagrelor significantly reduced platelet adhesion on atherosclerotic plaque, but not the platelet surface area (Figure 2B). Combining aspirin with R406 did not significantly reduce platelet adhesion compared to aspirin alone. Both platelet spreading and adhesion on collagen, fibrinogen, and fibrin were significantly inhibited by 10 µM of R406, which was significantly greater than the effect of aspirin alone. However, the effects were again concentration-dependent and a lower concentration of R406 (1 µM) had no significant effect on the platelet surface area or platelet adhesion on collagen, fibrinogen, or fibrin. The combination of aspirin and 10 µM of R406 caused a greater reduction in platelet spreading and adhesion on collagen, fibrinogen, and fibrin than the combination of aspirin and ticagrelor (Figure 2C,D).

### 2.3. R406 Inhibits Atherosclerotic Plaque-Induced Platelet Aggregation

Atherosclerotic plaque rupture has a critical role in ACS by acting as a potent stimulus for thrombus formation through the activation of platelet receptor GPVI [24,25]. We investigated the effect of R406 on atherosclerotic plaque-induced platelet aggregation in washed platelets, PRP, and whole blood to assess the effect of R406 in the presence and absence of plasma protein-binding. At a concentration of 3 µM, R406 significantly inhibited platelet aggregation in response to 70 µg/mL of plaque homogenate, and at a concentration of 10 µM, R406 completely blocked platelet aggregation in washed platelets (Figure 3A). The aggregation of washed platelets in response to plaque was also reduced by aspirin, but the addition of 10 µM of R406 caused significant additional inhibition compared to aspirin alone (Figure 3A). However, no effect on platelet aggregation was observed with R406 (10 µM) in PRP, even when in combination with both aspirin and ticagrelor, which significantly reduced platelet aggregation (Figure 3B). R406 also had no effect on platelet aggregation in response to atherosclerotic plaque in whole blood (Appendix A).

### 2.4. When Combined with Aspirin and Ticagrelor, R406 Provides Mild Additional Inhibition of Platelet Adhesion and Thrombus Formation over Atherosclerotic Plaque under Flow Conditions

We next investigated the effect of R406 alone and in combination with aspirin (30 µM) and ticagrelor (1 µM) on platelet aggregate formation over atherosclerotic plaque homogenate under arterial flow conditions. Healthy donor whole blood was incubated with R406, aspirin, and/or ticagrelor for 10 min at 37 °C and perfused over plaque homogenate at arterial shear rates (1000 s^−1^). R406 alone had no effect on platelet aggregate formation or platelet adhesion (Figure 4). Aspirin alone or combined with ticagrelor significantly reduced both platelet adhesion (surface area coverage) and aggregate formation (fluorescence intensity) on atherosclerotic plaque homogenate. Meanwhile, ticagrelor alone only reduced aggregate formation over atherosclerotic plaque. The addition of 10 µM of R406 to aspirin caused a slight but significant additional reduction of platelet adhesion compared to aspirin alone. R406 with ticagrelor showed a marginal reduction of platelet adhesion and aggregate formation compared to ticagrelor. Meanwhile, adding 10 µM of R406 to aspirin and ticagrelor provided modest additional inhibition (Figure 4).

### 2.5. Effect of R406 on GPVI-Mediated Platelet Function Is Amplified by Aspirin and Ticagrelor

We next confirmed that R406 inhibited other GPVI agonists and determined whether the effect of R406 on GPVI-mediated platelet function could be amplified with the addition of aspirin (30 µM) or ticagrelor (1 µM). In PRP, where the effective concentration of R406 is reduced by plasma protein binding [22], the inhibition of collagen-induced aggregation by R406 was concentration-dependent and could be overcome by increasing the agonist concentration. R406 (10 µM) completely abolished and partially inhibited platelet aggregation in response to 1 and 2.5 µg/mL of collagen, respectively. However, no inhibitory effect was observed at collagen concentrations above 2.5 µg/mL (Figure 5A). R406 was notably more potent in the absence of plasma proteins in washed platelets (Appendix A).

Both aspirin and/or ticagrelor showed partial inhibition of collagen-induced platelet aggregation in PRP compared to the vehicle control, with a 50% greater inhibition when combined compared to either alone. No additional inhibitory effect was observed with aspirin and ticagrelor in combination with 0.1–3 µM of R406. However, 10 µM of R406, which showed modest effects alone, demonstrated more potent inhibition in combination with these routinely used antiplatelet agents (Figure 5B). Similar results were observed in whole blood aggregometry, with R406 only showing inhibition at 10 µM, and greater inhibition in combination with aspirin and/or ticagrelor (Appendix A).

CRP-induced aggregation (3 µg/mL) was similarly inhibited by combining 10 µM of R406 with aspirin and/or ticagrelor, despite aspirin and/or ticagrelor having no effect and R406 only exhibiting partial inhibition at 10 µM (Figure 5C). The effect of R406 on CLEC-2-mediated platelet aggregation with the CLEC-2 agonist rhodocytin was also assessed. Significant inhibition in response to different concentrations of rhodocytin was similarly achieved at 10 µM of R406 (Figure 5D). Additional aggregation traces are shown in the Appendix A.

### 2.6. R406 Inhibits Platelet Aggregate Formation and Platelet Adhesion on Collagen under Flow Conditions When Added Ex Vivo to the Blood of Patients Treated with Aspirin and Ticagrelor

To further investigate the additive effects of R406 in combination with the widely used antiplatelet agents, aspirin and ticagrelor, we assessed platelet aggregate formation (fluorescence intensity) and platelet adhesion (platelet coverage) under arterial flow conditions (1000 s^−1^) over immobilised collagen. This was performed using whole blood from patients with ACS taking aspirin 75 mg once daily and ticagrelor 90 mg twice daily (DAPT), with R406 incubated ex vivo with blood for 10 min at 37 °C. The clinical characteristics of the patients involved in the study are provided in the Appendix A. When R406 was added to the blood from DAPT-treated patients, significant additive inhibition of both platelet adhesion and platelet aggregate development was observed with both 1 µM and 10 µM of R406 (Figure 6). The addition of R406 to PRP and whole blood from patients treated with aspirin and ticagrelor did not result in the further inhibition of platelet activation with the G protein-coupled receptor agonists TRAP, ADP, and arachidonic acid (ASPi) (Appendix A).

The ex vivo addition of either 10 µM of R406, 30 µM of aspirin, or 1 µM of ticagrelor to blood from healthy donors significantly inhibited platelet aggregate formation but not platelet adhesion on collagen under arterial shear (1000 s^−1^) (Appendix A, Appendix AA,B). Meanwhile, the combination of aspirin and ticagrelor significantly reduced both platelet adhesion and platelet aggregate formation (Appendix A). However, the addition of R406 to these conventional antiplatelet agents did not further amplify their effect except with aspirin, where a slight reduction in platelet adhesion was observed (Appendix A).

### 2.7. Fostamatinib Does Not Affect Arterial or Venous Thrombus Formation in Mice

To determine the effect of fostamatinib on arterial thrombosis, ferric chloride (FeCl_3_)-induced thrombus formation in the carotid artery was assessed. Thrombus formation was unaltered in fostamatinib-treated (80 mg/kg, gavage) mice compared to vehicle-treated controls (Figure 7A). Combination therapy with ticagrelor (30 mg/kg) and aspirin (25 mg/kg) was also assessed, with thrombus formation only reduced in the presence of ticagrelor (Figure 7B). The analysis of plasma collected from the mice following FeCl_3_-induced thrombosis indicated that reasonably high levels of the active metabolite (Appendix A) were present. To determine what concentration was needed for the complete inhibition of GPVI, we performed a dose-response of the effect of R406 on CRP-induced platelet P-selectin expression in mouse blood (data not shown). The in vitro addition of R406 to mouse blood indicated that higher concentrations than those achieved with in vivo dosing were required for the complete inhibition of GPVI activation (Figure 7C).

It has been suggested that the lower thrombosis rates observed in fostamatinib-treated ITP patients may in part be due to its inhibition of platelet CLEC-2, which signals via the same pathway as GPVI [26,27]. CLEC-2 has been shown to have a critical role in deep vein thrombosis formation in mice through interaction with its ligand podoplanin, with post-mortem data suggesting a similar mechanism may occur in humans [28]. To determine the effect of fostamatinib on deep vein thrombosis, we performed an inferior vena cava (IVC) stenosis model in mice. Fostamatinib was administered to mice via a formulated diet, with IVC ligation performed at day 6 and the thrombi assessed 48 h later. Thrombi formed in seven out of ten (70%) fostamatinib-treated mice and eight out of nine (90%) control mice (Figure 7D), which is within the expected prevalence range for this model [28,29]. Fostamatinib also had no effect on the size of the thrombi formed (Figure 7D). The analysis of the blood collected from these mice demonstrated reduced platelet activation in response to GPVI (CRP) stimulation but an unaltered response to CLEC-2 (rhodocytin) stimulation in fostamatinib-treated mice, explaining the lack of effect observed (Figure 7E).

## 3. Discussion

We explored whether the Syk inhibitor fostamatinib has the potential to be repurposed as an antiplatelet treatment for atherothrombosis. The main findings of this study were: (1) R406 mildly inhibited platelet responses induced by atherosclerotic plaque homogenate, likely due to GPVI inhibition; (2) anti-GPVI effects of R406 were amplified by the commonly-used antiplatelet medications aspirin and ticagrelor; (3) however, the effects of R406 were concentration-dependent and diminished in the presence of plasma proteins, which may explain why fostamatinib did not significantly inhibit thrombosis in murine models.

To our knowledge, this study was the first to examine the effect of R406 on atherosclerotic plaque-induced platelet responses and to comprehensively study the effects of R406 in combination with aspirin and ticagrelor, which are common treatments for atherothrombosis. R406 mildly inhibited atherosclerotic plaque-induced platelet activation, aggregation, and adhesion. Atherosclerotic plaque induced a pattern of signalling that was consistent with platelet GPVI activation, indicated by the phosphorylation of Syk Y525/526, LAT Y200, and PLC*γ*2 Y1217. The phosphorylation of each of these in response to plaque was fully inhibited by higher concentrations of R406, suggesting that R406 inhibits the atherosclerotic plaque-induced activation of GPVI. In keeping with a GPVI-mediated effect, R406 also inhibited other GPVI agonists, such as collagen and the specific GPVI agonist CRP, as has previously been shown [30,31]. We also confirmed that R406 inhibited CLEC-2, which is mediated by Syk, but had no effect on G protein-coupled receptors, such as P2Y_12_ or PAR1, which are not mediated by Syk. Taken together, these findings suggest that R406 mildly inhibited the GPVI-mediated activation of platelets by atherosclerotic plaque without any apparent off-target effects on G protein-coupled receptor-mediated pathways.

Aspirin and P2Y_12_ inhibitors, such as clopidogrel, prasugrel, and ticagrelor have well-established benefits in the treatment of ACS. Novel antiplatelet treatments for ACS would initially have to be combined with one or more of these medications, as it would be considered unethical to randomise a patient to not receive these in a clinical trial. Therefore, we determined whether R406 provides additional antiplatelet effects when combined with aspirin and/or the P2Y_12_ inhibitor ticagrelor. The effects of R406 on GPVI-mediated platelet function were largely additive with the effects of aspirin and ticagrelor. Notably, the concomitant use of aspirin or ticagrelor was required for the inhibition of platelet aggregation by R406. Aspirin and ticagrelor inhibit the secondary mediators of platelet activation that amplify and sustain platelet aggregation, mediated by thromboxane A2 and ADP, respectively [32]. Our findings suggest that the partial blockade of GPVI by R406 was insufficient to prevent the release of secondary mediators, which were then able to trigger complete platelet activation and aggregation.

In several assays, a 10 µM concentration of R406 was required to fully inhibit GPVI-mediated platelet function in the presence of plasma proteins in PRP. R406 was able to fully inhibit at lower concentrations and was more potent in the absence of plasma proteins in washed platelets, related to the high (~99%) plasma protein binding of R406. A 10 µM concentration of R406 may be supratherapeutic as the reported Cmax ranges between 0.8 and 2.5 µM [22]. However, less plasma protein binding and higher bioavailability of drugs may occur in patients compared to healthy volunteers, due to age, reduced first-pass clearance, reduced renal function, and drug–drug interactions [33]. In keeping with this, even a low concentration of R406 (1 µM) provided significant additional inhibition of platelet adhesion to collagen under flow conditions when added to the blood of patients with ACS treated with aspirin and ticagrelor. As this was the most representative model of atherothrombosis that we used, this suggests possible mild additional antithrombotic effects of R406 at therapeutic concentrations.

While the present findings using collagen, rhodocytin, and fibrinogen broadly agree with previous studies, there are differences in the concentration of R406 needed to mediate these effects among studies in the literature and compared to our study [30,31,34]. Spalton et al. reported the inhibition of downstream phosphorylation in response to collagen and CRP with 1 µM of R406, whereas, in the present study, 10 μM of R406 was required to block signalling [30]. In both studies, Syk inhibition could be overcome by increasing agonist concentrations. Differences in inhibitor concentrations required for blockade could therefore be explained by differing concentrations of collagen and CRP used in the studies. We also found only a mild effect of R406 on thrombus formation over collagen compared to previous reports of strong inhibition by Tullemans et al. [34]. This is likely due to the 3.3× higher concentration of R406 (33 µM) used by Tullemans et al. At these concentrations, R406 likely causes the off-target inhibition of kinases other than Syk [35,36], the most important of which are Src family kinases, which have important roles in a wide range of platelet functions [35,36]. Indeed, the Src inhibitor dasatinib is known to cause platelet dysfunction and bleeding independent of thrombocytopenia [37]. In the present study, we have also shown that R406 at a high concentration of 10 µM in washed platelets (where there is no plasma protein binding) also caused the complete blockade of upstream Src phosphorylation. This is likely to have also contributed to the effects of GPVI-mediated signalling. While Tullemans et al. also examined the effect of dasatinib on collagen-induced thrombus formation, finding less inhibition than R406, they did so at a concentration 330× lower (0.1 µM) than that of R406. Another flow study using higher concentrations of dasatinib reported stronger and more consistent inhibition of collagen thrombus formation by dasatinib (10 µM), and modest inhibition by R406 (10 µM), similar to our findings [38].

Mice deficient in GPVI and CLEC-2 exhibited a severe reduction in thrombosis and increased bleeding, with a similar phenotype reported in Syk-deficient mice [20,39,40,41]. The pharmacological inhibition of Syk (fostamatinib, PRT060318, and BI1002494), however, does not cause bleeding in mice, which was suggested to be due to the signalling-independent functions of GPVI and CLEC-2 in haemostasis and the differing order of SFK and Syk downstream of the receptors [16,18,19,20]. Protection against ferric chloride-induced arterial thrombosis and collagen- and epinephrine-induced venous thromboembolism has been shown with PRT030618 [18], while BI1002494 showed protection against aortic crush/mechanical injury of abdominal aorta-induced thrombosis [20]. However, we did not see an effect of fostamatinib in our ferric chloride injury model of arterial thrombosis, despite showing that fostamatinib partially inhibited GPVI (shown by reduced CRP-induced P-selectin expression). Differences in the vessel size and rheology (mesentery vs. carotid), experimental measurement (occlusion time vs. thrombus fluorescence), as well as inhibitor selectivity and dosing strategy (continuous perfusion vs. oral gavage) are potential explanations for these differing findings with the multifaceted ferric chloride injury model [42]. Fostamatinib is rapidly cleared in mice with a half-life of approximately 30 min (personal communication with Rigel Pharmaceuticals) [43]. Although we performed the experiments immediately after dosing and used a high dose that achieved higher levels of active metabolite than those observed in humans, it does not seem to have been enough to completely inhibit GPVI in mice. Indeed, Van Eeuwijk et al. reported limited efficacy in mice with commercial Syk inhibitors (they did not specify which), before using BI1002494 and showing protection in the highly GPVI-mediated aortic crush/mechanical injury of abdominal aorta model of thrombosis [20,44]. Similarly, a lack of protection from venous thrombosis in mice could be explained by the incomplete inhibition of CLEC-2 by fostamatinib. In addition, CLEC-2 supports thrombus formation as an adhesion receptor [45]. Previous reports of PRT060318 protection against venous thromboembolism may be due to the direct inhibition of the collagen (and epinephrine) used to induce thrombosis [18].

The present study provides insights into a recent 5-year follow-up study of fostamatinib, which was associated with low levels of thromboembolic events in patients with ITP, who are known to have increased risk of thrombosis despite thrombocytopenia [27]. Our findings suggest mild antithrombotic effects of fostamatinib may have contributed to this finding. Reassuringly, there was also a reduced incidence of bleeding in these patients, possibly related to increasing the platelet count by the inhibition of Fc*γ* receptor (Fc*γ*R)-mediated platelet phagocytosis by macrophages [46]. Indeed, pharmacological Syk inhibition is well-tolerated in patients and does not appear to be associated with increased bleeding [21,47,48]. This lack of bleeding, even in thrombocytopenic patients, suggests that Syk inhibition does not affect platelet function in haemostasis, in which multiple Syk-independent platelet activation pathways are critically involved [30,49,50].

## 4. Materials and Methods

### 4.1. Reagents

The details of the sources of reagents, methods, and analyses can be found in the online Appendix A.

### 4.2. Platelet Function Tests

Platelet function studies were performed using the Platelet Aggregation Profiler (PAP-8E) aggregometer (Bio/Data Corporation, Horsham, PA, USA), Multiplate^®^ analyzer (Roche Diagnostics, Munich, Germany), the Optimul platelet aggregation assay (Queen Mary University, London, UK) [51], and a microfluidic flow adhesion assay. Optical assay to assess platelet spreading was analysed using a validated analysis method [52]. Rhodocytin was isolated according to a published protocol and provided by Johannes Eble from the University of Münster [53]. The phosphorylation of downstream signalling proteins was assessed by Western blot. Throughout the study, a concentration of 30 µM of aspirin and 1 µM of ticagrelor were chosen since these were the concentrations that provided complete inhibition of their target receptors (Appendix A). Since washing platelets desensitises platelets to ADP, we also investigated the effects of R406, aspirin, and ticagrelor in platelet-rich plasma and whole blood.

### 4.3. Blood Collection and Ethical Approval

Blood was collected from drug-free healthy, consenting volunteers in accordance with the Declaration of Helsinki and with ethical approval granted by the University of Birmingham internal ethical review board (ERN-11-0175), and from patients that were treated with either aspirin or the combination of aspirin and ticagrelor (ethical approval from the National Research Ethics Service 18/WM/0386, study identifier IRAS 238646). Healthy volunteer samples were collected by venepuncture in 4% sodium citrate-filled syringes. Patient samples were collected by venepuncture into trisodium citrate buffer solution (0.106 mol/L; S-Monovette 10 mL 9NC; Sarstedt, Nümbrecht, Germany) or Hirudin (525 ATU Hirudin/ml blood; S-Monovette 1.6 mL; Sarstedt, Nümbrecht, Germany).

### 4.4. Atherosclerotic Plaque Preparation

Atherosclerotic plaque was collected from 10 patients with symptomatic carotid artery stenosis undergoing carotid endarterectomy. The plaque samples were homogenised, pooled, and resuspended in phosphate-buffered saline (PBS), before being frozen at −80 °C. The protein concentration in the supernatant was determined using a standard bicinchoninic acid (BCA) protein assay kit (Thermo Fisher Scientific, Gloucester, UK). The dose response comparing atherosclerotic plaque to collagen and CRP-induced platelet aggregation in PRP using light transmission aggregometer (LTA) are provided in the Appendix A.

### 4.5. Animals

All animal experiments were performed using wild-type mice on a C57BL/6 background obtained from Charles River, United Kingdom. All procedures were in accordance with the Animal (Scientific Procedures) Act 1986 and undertaken with United Kingdom Home Office approval.

Concentrations of R406 active metabolite were measured in mouse plasma following oral gavage of mice with 80 mg/kg fostamatinib in vehicle or in vehicle plus DMSO (2%). Plasma samples were collected 1-hour post oral dosing and R406 metabolite measured as previous [54].

### 4.6. In Vivo Thrombosis

Murine models of arterial thrombosis (ferric chloride-induced injury of carotid artery) and deep vein thrombosis (inferior vena cava stenosis with partial flow restriction) were performed as previously described [28,55,56].

### 4.7. Platelet Preparation

Platelet-rich plasma (PRP) was generated from citrate-anticoagulated human blood by centrifugation. Washed human platelets were obtained by the further centrifugation of PRP in the presence of prostacyclin and resuspended in modified Tyrode’s buffer.

### 4.8. Statistical Analysis

All results are presented as the mean ± standard error of the mean (SEM). Unless otherwise stated, the statistical analyses were performed using one- or two-way ANOVA with Dunnett’s correction for multiple comparisons. Mixed model analysis was used if data values were missing. A *p* value of < 0.05 was considered to be statistically significant. All analyses were performed using GraphPad Prism software version 9.2 (GraphPad, San Diego, CA, USA).

## 5. Conclusions

In conclusion, the active metabolite (R406) of the Syk inhibitor fostamatinib provides modest inhibition of platelet responses induced by atherosclerotic plaque and other GPVI agonists, which is amplified by conventional antiplatelet agents such as aspirin and ticagrelor. Although the effects appeared to be mild and were insufficient to prevent thrombosis in animal models, they may have contributed to the low rates of thrombosis seen in patients treated with fostamatinib for ITP.

## Figures and Tables

**Figure 1 ijms-23-06982-f001:**
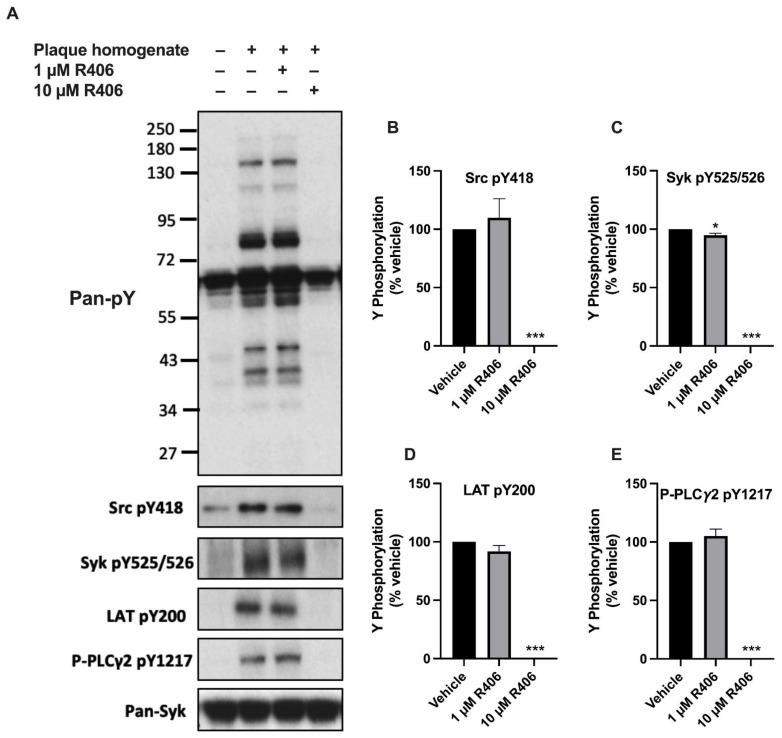
Phosphorylation of R406 blocks Src, spleen tyrosine kinase (Syk), linker for activation of T cells (LAT), and PLC*γ*2 induced by atherosclerotic plaque homogenate. Healthy donor washed human platelets (4 × 10^8^/mL) were incubated with R406 or vehicle (0.1% DMSO) for 10 min, then stimulated with atherosclerotic plaque homogenate (70 µg/mL) for 3 min in the presence of eptifibatide (9 μM). Platelets were then lysed, separated with SDS-PAGE, and Western blotted for tyrosine phosphorylation (pY). (**A**) Representative images of 3 independent experiments. Densitometry quantification of (**B**) Src pY418 (**C**) Syk pY525/526 (**D**) LAT pY200, and (**E**) PLC*γ*2 pY1217. Band intensities were normalised to vehicle control. Results are shown as mean ± SEM. One-way ANOVA with Dunnett’s correction for multiple comparisons, * *p* < 0.05, *** *p* < 0.001.

**Figure 2 ijms-23-06982-f002:**
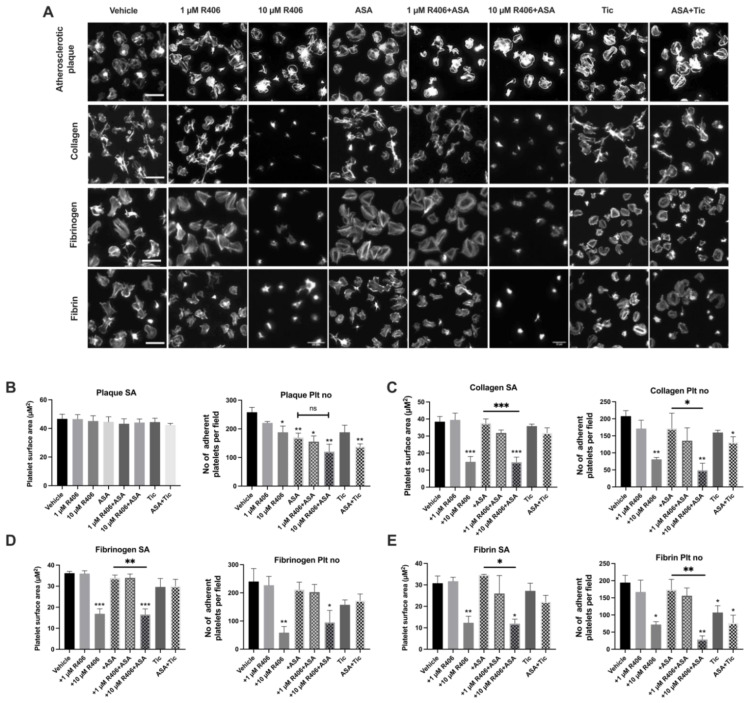
Effect of R406 on platelet spreading and adhesion on multiple ligands. (**A**) Representative images of spread platelets on atherosclerotic plaque material. Healthy donor washed human platelets (2 × 10^7^/mL) were incubated with either vehicle, R406, and/or 30 µM of aspirin for 30 min at 37 °C, and then allowed to spread for 30 min on (**B**) atherosclerotic plaque (70 µg/mL) (**C**) Horm collagen (10 μg/mL), (**D**) fibrinogen (10 μg/mL), or (**E**) fibrin (10 μg/mL). Platelets were then fixed and permeabilised. Actin was stained with phalloidin AF488 and surface area was quantified using a semi-automated machine learning workflow. Mean platelet surface area (SA) quantification and mean number of adherent platelets per 44,648 µm^2^. Results are shown as mean ± SEM. Results compared to vehicle control (above bars) or aspirin (indicated with lines) for R406 + ASA using one-way ANOVA, with Dunnett’s correction for multiple comparisons, *n* = 3–4. * *p* < 0.05, ** *p* < 0.01, *** *p* < 0.001. Scale bar = 10 μm. ASA, aspirin; Tic, ticagrelor.

**Figure 3 ijms-23-06982-f003:**
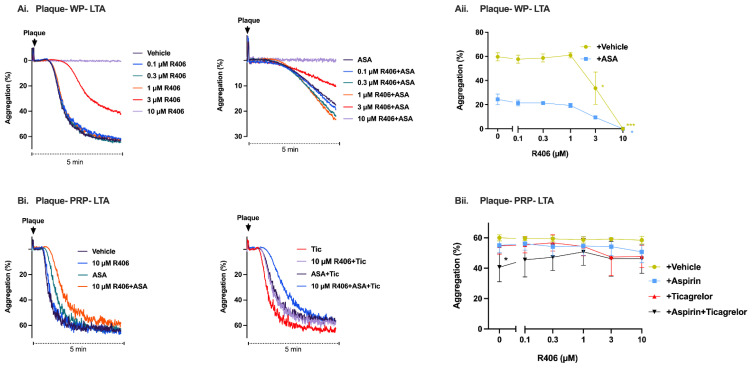
R406 inhibited atherosclerotic plaque-induced platelet aggregation in washed platelets but not PRP or whole blood. Platelet aggregation in healthy donor (**A**) washed platelets (**B**) PRP assessed by light transmission aggregometer (LTA). Samples were incubated with either vehicle, R406, 30 µM of aspirin or 1 µM of ticagrelor for 10 min, then stimulated with 70 μg/mL of plaque homogenate. Additional representative traces in Appendix A, Appendix A. Results are shown as mean ± SEM of three independent experiments. Results analysed with two-way ANOVA, with Dunnett’s correction for multiple comparisons vs. vehicle control or vs. corresponding comparator (aspirin and/or ticagrelor alone) (* *p* < 0.05, *** *p* < 0.001). WP, washed platelets. ASA, aspirin. Tic, ticagrelor.

**Figure 4 ijms-23-06982-f004:**
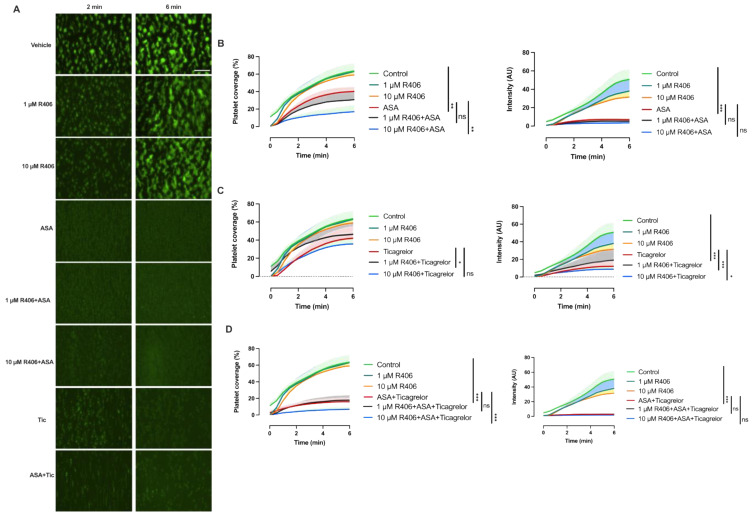
R406 does not inhibit platelet adhesion over atherosclerotic plaque. Healthy donor whole blood was incubated for 10 min at 37 °C with R406, 30 µM of aspirin, and/or 1 µM of ticagrelor, and then perfused over 1 mg/mL plaque homogenate-coated chambers at a shear rate of 1000 s^−1^. Platelets were labelled with 2 µM of DiOC_6_ for 10 min prior to perfusion for visualisation. (**A**) Representative images display platelet aggregate formation at 2 and 6 min after the start of perfusion. Scale bar = 100 µm. Quantification of platelet surface area coverage and platelet aggregate size (fluorescence intensity) in (**B**) R406 ± aspirin, (**C**) R406 ± ticagrelor, and (**D**) R406 ± aspirin+ticagrelor-treated blood. Measurements were taken every 30 s. Mean (solid line) + SEM (shaded area); *n* = 4. Statistical comparisons were made at 6 min compared to vehicle control or aspirin and/or ticagrelor after adding R406 using repeated measures ANOVA or, if inappropriate, a mixed-effect model with Dunnett’s correction for multiple comparisons. * *p* < 0.05, ** *p* < 0.01, *** *p* < 0.001. ASA, aspirin. Tic, ticagrelor. ns = non-significant.

**Figure 5 ijms-23-06982-f005:**
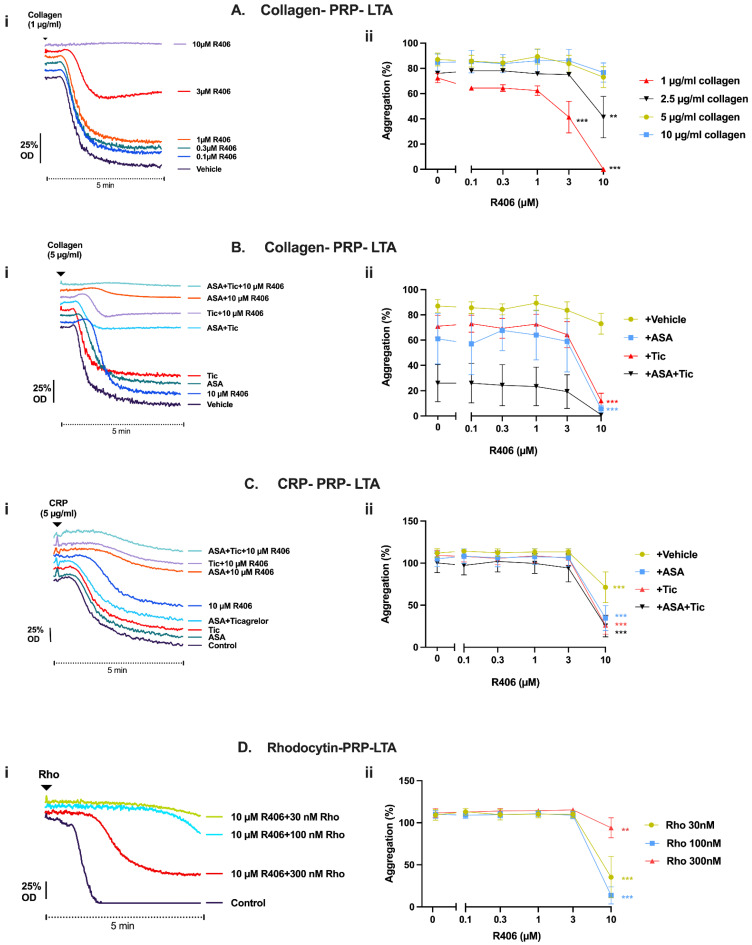
Effect of R406 in combination with aspirin and/or ticagrelor on collagen- and CRP-mediated platelet aggregation and effect of R406 on rhodocytin-mediated platelet aggregation. (**A**–**C**) GPVI- and (**D**) CLEC-2-mediated platelet aggregation in healthy donor PRP assessed with LTA. Samples were incubated with either vehicle (0.1% DMSO), R406, 30 µM of aspirin and/or 1 µM of ticagrelor for 10 min, then stimulated with (**A**) 1–10 μg/mL of collagen, (**B**) 5 μg/mL of collagen, (**C**) 3 µg/mL of CRP, or (**D**) 30–300 nM of rhodocytin to induce aggregation. (i) Representative platelet aggregation traces. Full sets of traces are provided in Appendix A and Appendix A. Results are shown as mean ± SEM of three independent experiments. Results analysed with two-ANOVA with Dunnett’s correction for multiple comparisons vs. vehicle control. ** *p* < 0.01, *** *p* < 0.001. ASA, aspirin. Tic, ticagrelor, Conc, concentration.

**Figure 6 ijms-23-06982-f006:**
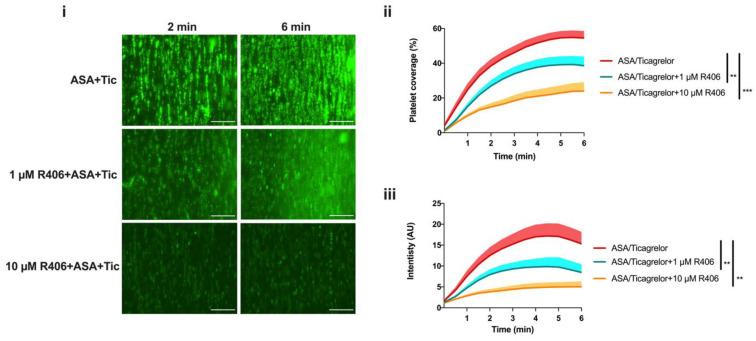
R406 provided additive inhibition of platelet thrombus formation over collagen when added to blood of patients taking aspirin and ticagrelor dual antiplatelet therapy. Whole blood from patients taking aspirin and ticagrelor was incubated with R406 or vehicle (0.1% DMSO) for 10 min at 37 °C, then perfused over a 200 µg/mL Horm collagen-coated chamber at a shear rate of 1000 s^−1^. Platelets were labelled with 2 µM of DiOC_6_ for 10 min prior to perfusion for visualisation. (**i**) Representative images display platelet aggregate formation at 2 and 6 min after start of perfusion. Scale bar = 100 µm. (**ii**) Quantification of platelet surface area coverage and (**iii**) platelet aggregate size (fluorescence intensity). Measurements were taken every 30 s. Mean (solid line) + SEM (shaded area); ASA/Ticagrelor, *n* = 12; ASA/Ticagrelor + 1 µM of R406, *n* = 11; ASA/Ticagrelor + 10 µM of R406, *n* = 6. Statistical comparisons were made at 6 min using repeated measures ANOVA or a mixed-effect model with Dunnett’s correction for multiple comparisons. ** *p* < 0.01, *** *p* < 0.001. ASA, aspirin. Tic, ticagrelor.

**Figure 7 ijms-23-06982-f007:**
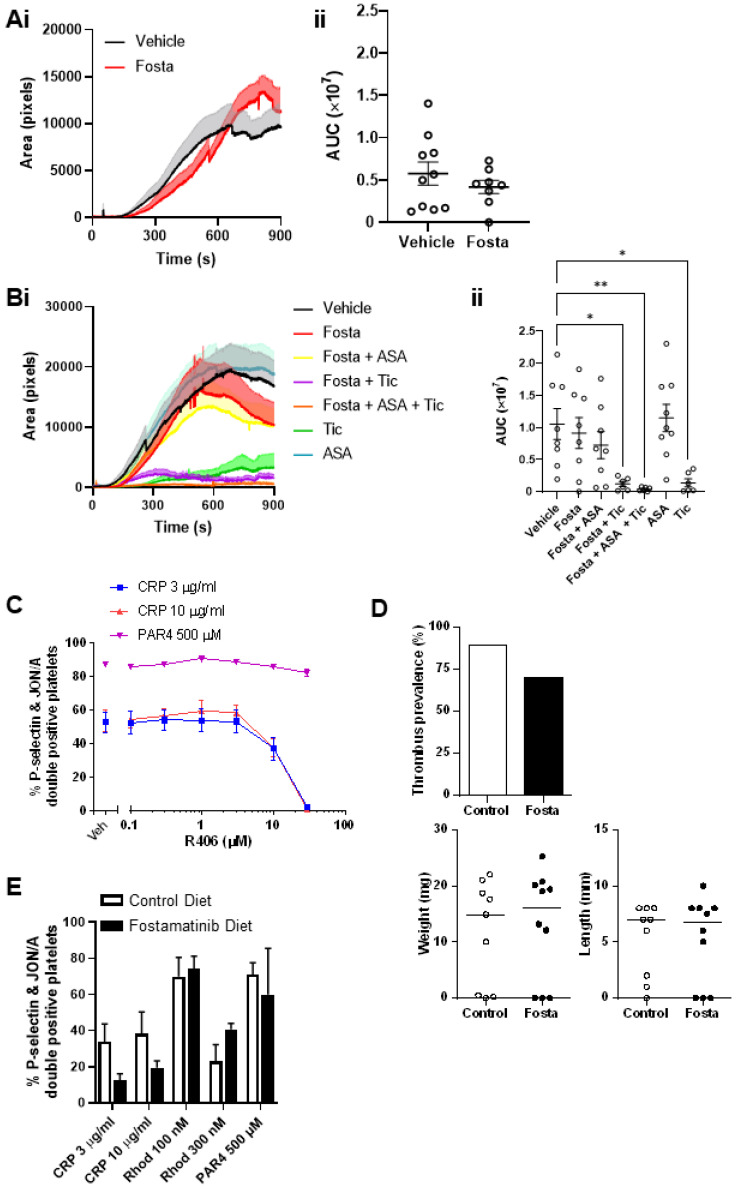
Fostamatinib does not affect arterial or venous thrombus formation in mice. Ferric chloride-induced thrombosis in wild-type mice orally dosed with (**A**) fostamatinib (80 mg/kg) or DMSO vehicle or (**B**) fostamatinib (80 mg/kg) in combination with aspirin (25 mg/kg) and ticagrelor (30 mg/kg). Following oral gavage, the mice were anaesthetised, carotid arteries exposed and injured with 10% ferric chloride for 3 min. Mice were injected with DyLight488-conjugated anti-GPIbβ antibody (0.1 μg/g body weight, X488, Emfret Analytics) prior to injury, and the accumulation of labelled platelets into the thrombi was assessed. (**i**) Each curve represents the mean ± SEM thrombus area in pixels. (**ii**) Area under the curve quantification of the thrombus area (mean ± SEM; *n* = 8–10 mice), with comparisons assessed by Mann–Whitney test. * *p* < 0.05, ** *p* < 0.01 (**C**) Mouse blood was incubated with R406 for 10 min, then stimulated with CRP or PAR4 peptide. Surface expression of P-selectin and activated integrin αIIbβ3 (JON/A) measured by flow cytometry (mean ± SEM; *n* = 3). (**D**) Wild-type mice were fed fostamatinib-formulated or control diet for 6 days before IVC stenosis surgery was performed. The mice were then allowed to recover, and thrombi were collected 48 h later (day 8). Thrombus prevalence and weight and length are shown (*n* = 9–10). Horizontal line represents the median. Statistical analysis on thrombus prevalence was done with Fisher’s exact test. Statistical analysis for thrombus weight and length was performed with Mann–Whitney test. (**E**) Mouse blood was collected from fostamatinib- and control diet-treated mice, stimulated with CRP, rhodocytin, or PAR4 peptide. Surface expression of P-selectin and activated integrin αIIbβ3 (JON/A) measured by flow cytometry (mean ± SEM; *n* = 3–5 mice). Fosta, fostamatinib. ASA, aspirin. Tic, ticagrelor. Rhod, rhodocytin.

## Data Availability

Not applicable.

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
