# Peer review of "Antithrombotic Effects of Fostamatinib in Combination with Conventional Antiplatelet Drugs"

_ijms, 2022, doi:10.3390/ijms23136982_

Round 1

Reviewer 1 Report

The study by Harbi and Smith et al., attempts to identify antithrombotic, platelet-mediated effects of the syk inhibitor fostamatinib (R406 = active metabolite) alone and in combination with duel antiplatelet therapy in a series of in vitro, ex vivo and in vivo experiments. The experimental work is competent, and the manuscript is clear and well written, especially given the mixed nature of the results. The authors provide some evidence of for an antithrombotic effect of R406 in samples from ACS patients receiving DAPT, but weaker evidence in in vitro studies of drug-free healthy donors. The murine studies identify no anti-thrombotic properties, although as the authors point out, this may have more to do with limitations of the murine model. The mixture of positive and negative data makes the study difficult to interpret, but this should not necessarily be a barrier to publication. However, it is not entirely clear if some of the negative findings might be due to the experimental conditions utilised in the study.

1   1.       Figure 1. Fostamatinib is a dirty drug which inhibits many platelet kinases besides Syk. It is important to understand if the inhibitory effects the authors observe at 10µM R406 in washed platelets are due to inhibition of Syk or other kinases. The complete ablation of tyrosine phosphorylation observed at 10µM R406 suggest inhibition of other kinases at this concentration. Specifically, the authors should check if src family kinases are inhibited under these conditions (10µM R406, washed platelets). In the discussion the authors state “taken together, these findings suggest that Syk inhibition by R406 mildly inhibited GPVI-mediated activation of platelets by atheroscle-rotic plaque without any apparent off-target effects on platelets” but is not demonstrated empirically within the manuscript. The authors go on to discuss the possibility of R406 inhibiting src family kinases in other studies but do not provide evidence that this is not happening in their own study.

2    2.       Fig. 2. results are slightly surprising due to the minimal effects of ASA and ticagrelor and raise the following questions and issues:

a.       Is the concentration of plaque used in this assay too high (supramaximal)? – This might mask inhibitory effects. Have the authors tried this assay with lower concentrations of plaque? It is difficult for the reader to understand the relative strength of 70µg/ml plaque compared to the other coatings used in these experiments due to the undefined nature of plaque homogenate. The authors should repeat this assay with lower concentrations of plaque.

b.       The authors’ protocol for preparing washed platelets does not include steps required to avoid desensitisation of ADP receptors (such as inclusion of apyrase). This may explain why ticagrelor exerts only mild inhibitory effects in this assay, as the platelets are already largely unresponsive to ADP. The rest of the study utilises whole blood or PRP, in which platelets are likely to be sensitive to ADP. To investigate the effects of ticagrelor and make this figure comparable to the rest of their study, this assay could be repeated with washed platelets prepared using a protocol designed to preserve ADP sensitivity.

3    3.       Figure 3B raises questions regarding the strength of the 70µg/ml plaque as a stimulus as it appears almost insurmountable by ASA and ticagrelor in the aggregation assay. The authors could include a concentration-response curve for the homogenate (without inhibitors). If this indicates that plaque is being a used at a supramaximal concentration, the authors should repeat the PRP aggregation assays with lower concentrations of plaque.

 4.       Figure 4: ASA and ticagrelor cause very strong inhibition on their own in the absence of R406. Additive inhibition caused by R406 would therefore be v. difficult to distinguish. The concentrations of ASA and ticagrelor utilised in this assay probably represent full blockade of COX1/P2Y12, and therefore provide little scope for further inhibition by R406. Are the effects of R406 more pronounced at submaximal concentrations of ASA and ticagrelor that cause only partial inhibition?

   5.       The legends for Figure 5 Ai, Bi, Ci are difficult to match up with the lines. The legend for Di is incorrect (all lines are labelled 10uM R406). The legends for Bii and Cii are also confusing – The concentration of R406 is labelled on the x-axis but ‘R406’ is also included in the legend alongside other inhibitors that are included at fixed concentrations. Agonist concentrations in this figure again seem quite high. Are effects of R406 seen at lower concentrations if the agonist stimulation isn’t as strong?

6   6.       The legend text for figure S9 incorrectly states that the experiments were performed on plaque.

7   7.       Figure 6 is the strongest figure in the manuscript because it utilises samples from patients with ACS on DAPT and demonstrates the strongest effect of R406. It may provide support for the possibility that there is more scope for additive inhibition by R406 when inhibition of COX1 and P2Y12 are submaximal – as might be the case when ASA and ticagrelor are dosed orally rather than added to blood ex vivo. In the discussion, the authors suggest that the ACS patients may be more susceptible to inhibition by R406 for age and disease-related reasons and do not explore the possibility that inhibition by R406 might be more apparent in the presence of submaximal DAPT. It would strengthen the manuscript if this possibility was explored in in vitro thrombus formation assays on collagen (to match fig 6) using blood from healthy donors treated with lower concentrations of ASA and ticagrelor +/- R406.

Reviewer 2 Report

The manuscript presents an original study. It is well written and interesting. The authors evaluated fostamatinib, a Syk inhibitor, alone or in combination with the established antithrombotic therapy of ACS, to see if it can be used as antiplatelet treatment for atherothrombosis. The laboratory assessment is very complex, resulting in many data. It is to be appreciated that the authors presented the results in a very organized, clear, well-structured manner, with images and graphics that facilitate the reading. The results are new and take the stage of knowledge on this topic a step further.

In row 18 please add: The aim of our study was to...

Thank you!

Author Response

Response to Reviewer 2 comments

The manuscript presents an original study. It is well written and interesting. The authors evaluated fostamatinib, a Syk inhibitor, alone or in combination with the established antithrombotic therapy of ACS, to see if it can be used as antiplatelet treatment for atherothrombosis. The laboratory assessment is very complex, resulting in many data. It is to be appreciated that the authors presented the results in a very organized, clear, well-structured manner, with images and graphics that facilitate the reading. The results are new and take the stage of knowledge on this topic a step further.

In row 18 please add: “The aim of our study was to...”

Thank you very much for your review of out manuscript and appreciating that we have gone to lengths to ensure that our assessment of the work accurately reflects the complexity of the results. We have added the sentence that you suggest.

Reviewer 3 Report

This manuscript titled “Antithrombotic effects of fostamatinib in combination with conventional antiplatelet drugs” investigated whether ITP FDA approved drug fostamatinib can be repurposed as a as antithrombotic treatment.  Furthermore, this study assesses the impact of this drug on atherosclerotic plaque-mediated platelet responses alone and in combination with other known antiplatelet drugs namely Aspirin and ticagrelor which is currently in use for the prevention of thrombosis. This manuscript is interesting and well written. However, several issues needs to be addressed.

Major:

-           - The effect of R406 alone and in combination with other antiplatelet on hemostasis needs to be examined to address the gap in current antiplatelet therapy as indicated as one of the premises to conduct this study.

-          - The author needs to provide complete blood count including platelet count for both human subject and mice.

-           - For Atherosclerotic plaque preparation. There are reports (DOI: 10.1007/s11095-019-2619-2) in which they say that it’s not recommended to freeze in PBS, as there can be a drastic pH shifts that can affect proteins. It was not clear if freezing may impact the results.

-          - The authors have shown that R406 has a mild but insufficient prevention of thrombosis in animal model, It was not clear if the goal of this manuscript to repurpose fostamatinib as a monotherapy or in combination with conventional antiplatelet therapy.

-         -  It will add value to the manuscript to add a table with clinical characteristics of study participants (age, gender, etc.).

Minor

- Punctuation mistake on line 19 of the abstract.

-In supplemental figure 6, there’s a mistake on the footnote’s title.

-In figure 5D i, all four aggregation traces have the same conditions and it’s unclear if each color represents a different concentration of rhodocytin.

- Typo line 89 “Bicinchroninic Acid” to “Bicinchoninic Acid”

-Grammar mistake in line 360

- Line 300 typo “due to it inhibition” to “due to its inhibition”

- The quality of the spreading assay images needs some improvement  

Round 2

Reviewer 3 Report

Thank you for addressing all my comments.